# Modelling and Analysis of the Spital Branched Flexure-Hinge Adjustable-Stiffness Continuum Robot

**Nan Ma** , **Stephen Monk** and **David Cheneler** *

Department of Engineering, Lancaster University, Lancaster LA1 4YW, UK
* Correspondence: d.cheneler@lancaster.ac.uk

**Abstract:** Continuum robots are increasingly being used in industrial and medical applications due to their high number of degrees of freedom (DoF), large workspace and their ability to operate dexterously. However, the positional accuracy of conventional continuum robots with a backbone structure is usually low due to the low stiffness of the often-lengthy driving cables/tendons. Here, this problem has been solved by integrating additional mechanisms with adjustable stiffness within the continuum robot to improve its stiffness and mechanical performance, thus enabling it to be operated with high accuracy and large payloads. To support the prediction of the improved performance of the adjustable stiffness continuum robot, a kinetostatic model was developed by considering the generalized internal loads that are caused by the deformation of the flexure-hinge mechanism and the structural stiffening caused by the external loads on the end-effector. Finally, experiments were conducted on physical prototypes of 2-DoF and 6-DoF continuum robots to validate the model. It was found that the proposed kinetostatic model validates experimental observations within an average deviation of 9.1% and 6.2% for the 2-DoF and 6-DoF continuum robots, respectively. It was also found that the kinematic accuracy of the continuum robots can be improved by a factor of 32.8 by adding the adjustable stiffness mechanisms.

**Keywords:** continuum robot; adjustable stiffness; kinetostatic modelling; flexure hinge

## 1. Introduction

In recent years, continuum robots have increasingly been used for operations in confined and complex spaces due to their inherent dexterity, compact structure, large potential workspace, and high environmental adaptability [1–3]. However, most continuum robots are constructed using modular backbone structures and actuated by long driving cables/tendons with the actuators normally located remotely, far away from the area of operation of the manipulator [4]. The stiffness of a conventional continuum robot is an important factor as it affects its kinematic accuracy. This stiffness is mainly dependent on the tension of the driving cables (higher tension equates to higher stiffness [5]) and robot shape (i.e., bending angle and bending direction [6]). By just increasing the tension of the driving cables, the stiffness of the continuum robot can be enhanced to some extent, but this results in a change in the internal loads, especially in the friction along the backbone. Moreover, the high tension can result in a higher risk of creep in the driving cables, structural damage, and reduced stability of the system [7,8].

Significant efforts have been made to develop continuum robots to satisfy the requirements of specific industrial and medical applications. For example, researchers at Columbia University developed a multi-segment continuum robot (i.e., 17-DoF, diameter: 6.4 mm, length: 75 mm) [9], and later a dual-arm manipulator with 21 joints with a rigid 'central stem' structure [10] to perform single port access surgery. Similar backbone structures have been widely adopted to guide the continuum robot into deep/confined spaces for high-accuracy operations. However, the dexterity and accessibility of the continuum robot is limited by this structure [10]. Due to the slenderness of these structures, the payload

capability is often low. Researchers at Nottingham University recently developed a 16-DoF large length–diameter ratio continuum robot (length: 1.5 m, diameter: 12.6 mm) composed of ten 1-DoF sections and three 2-DoF sections, for inspection and repair in the confined space of aeroengines [11]. However, the stiffness of the system was largely limited by its slender structure.

Soft, steerable, 'snake' robots have been developed at Stanford University and the University of California in order to improve navigation in tightly constrained environments through the controlled asymmetric growth or extension of the robot body [12,13]. However, as the actuation is via pneumatic control of a highly compliant body, the payload capacity of the robot is quite low. Generally, it is found that the payload capacity for continuum robots with a slender structure or high dexterity is too low for most practical applications, whereas those with an acceptable payload capacity are often bulky structures with low dexterity.

Here, the mechanical performance (stiffness, payload capacity) of a continuum robot was improved with the inclusion of flexure hinge-based mechanisms. Flexure hinge-based mechanisms have been developed for many applications over the past few decades at the macro-scale (e.g., rotational joints [14] and parallel manipulators [15]) and micro-scale (e.g., scanning electron microscopy (SEM) [16] and cell injection [17]). Including flexure mechanisms in a system can greatly improve its kinematic accuracy by removing backlash and improve its working life by reducing creep in the cables and the load on actuators. For example, by using three radially symmetrically distributed cross-spring mechanisms, a parallel universal joint with constant rotational stiffness was developed [18]. Further, a variable stiffness actuator was realised wherein the stiffness can be adjusted through the changing of the second moment of area of the spring elements within, allowing for joint-stiffness control of robots [19]. By adopting four generalized cross-spring pivots, a 2-DoF rotational flexure mechanism was developed for precise unbalance measurements [20]. From the previous references, it can be seen by integrating flexure-hinge mechanisms within a continuum robot, the stiffness of the system can be actively controlled for specific applications, thus regulating the kinematic accuracy.

In addition, many studies have focused on kinematic [21,22] and stiffness [23,24] modelling of conventional backbone structured continuum robots in order to control them. However, if flexure hinges are to be integrated with the existing continuum arm, a new kinetostatic model needs to be derived in order to determine their mechanical performance, and further, to guide the structure and controller design [25]. Research has also been conducted on modelling the kinetostatic behaviour of flexure hinges [26,27]. For example, by utilizing the generalized equation for conic curves in polar coordinates, a 6-DoF compliance matrix was developed for flexure hinges with different conic sections [28]. Moreover, a 2-DoF compliant parallel universal joint was theoretically and physically proven to have constant rotational stiffness [18] when three identical cross-spring flexures were equally spaced around the central compliant shaft.

Kinetostatic modelling has also been adopted to predict the behaviour of continuum robots with integrated compliant mechanisms. For example, the kinetostatic equation was applied to a continuum robot with a flexible backbone to predict its deformation under external load [29], thus allowing the optimization of the structural design to improve its performance. The stiffness of a continuum robot that included compliant joint-based modular 2-DoF segments [30] was modelled to determine the displacement as a function of the externally applied load throughout the workspace to demonstrate its enhanced ability to access confined spaces.

Until now, to the best knowledge of the authors, most of the continuum robots that have incorporated additional compliant elements into their structure to enhance their stiffness have done so by modifying the compliance of the central shaft in the backbone. In contrast, the inclusion of flexure hinges (i.e., hinges constructed with compliant material with specific mechanical properties) within the modular structure of a continuum robot to improve the kinematic accuracy, and the corresponding kinetostatic modelling to predict its performance, have not been utilized. Here, the enhanced performance of a continuum

robot arm that incorporates a novel leaf spring-like flexure hinges is demonstrated. To address these challenges, this paper focuses on solving the following problems: (1) a new kinetostatic and stiffness model of the 6-DoF continuum arm (i.e., modular flexible hinges are incorporated) is developed to investigate the mechanical performance variation; and (2) a physical prototype and control system are developed to experimentally validate the proposed model.

## 2. Kinetostatic Model of the 6-DoF Continuum Robot

The compliance matrix of a continuum robot composed of serially connected 2-DoF segments with intermediate flexure hinges is established in this section. The flexure hinge was designed with a special structure to achieve the desired mechanical behaviour (i.e., a specific compressional stiffness while ensuring a much higher rotational stiffness). The compliance matrix for both a 2-DoF section and a continuum robot was derived in turn using Ryu's flexure mechanism modelling method [26], which is based on a Cartesian coordinate system located at the base of each hinge.

### 2.1. Stiffness Model of Beams within a Flexure Hinge

Each 2-DoF section of the continuum robot needs to achieve orthogonal rotational motion. To achieve the desired behaviour, each 2-DoF segment within the section includes three identical leaf spring-like flexure hinges that are equally radially spaced around central ball joints that connect disk $i$ and $i + 1$ respectively. As seen in Figure 1, these flexure hinges were designed to have a hexagonal shape with circular corners, to improve their load-bearing performance.

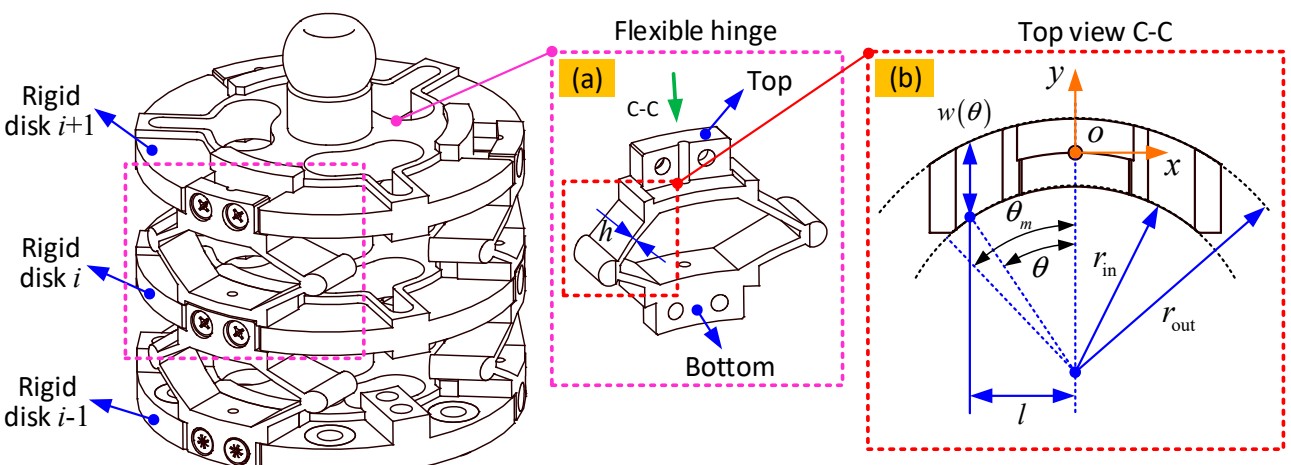

**Figure 1.** Structure of the modular 2-DoF segment, and parameter definition of the flexure hinges. Note: the enlarged view of (**a**) is the reverse of the original view of the mechanism on the left for better display, while (**b**) is the top view of the flexible hinge.

As the thickness of the beams that form the leaf-spring sections of the flexure hinge is much smaller than their length, their mechanical behaviour can be determined by Ryu's flexure mechanism modelling method [26]. In order to establish the compliance model of a whole segment, which is comprised of three flexure hinges, as shown in Figure 1, the 6-DoF compliance model of each single leaf spring, or beam, is first established. $h$ is the

thickness of the flexible hinge. This model considers the bending, torsion, axial and shear deformation of the beam under external loads, and can be expressed as:

$$\mathbf{C}_{\text{Leaf\_spring}} = \begin{bmatrix} \frac{dx}{dF_x} & 0 & 0 & 0 & 0 & 0 \\ 0 & \frac{dy}{dF_y} & 0 & 0 & 0 & \frac{d\theta_z}{dF_y} \\ 0 & 0 & \frac{dz}{dF_z} & 0 & \frac{d\theta_y}{dF_z} & 0 \\ 0 & 0 & 0 & \frac{d\theta_x}{dM_x} & 0 & 0 \\ 0 & 0 & \frac{d\theta_y}{dF_z} & 0 & \frac{d\theta_y}{dM_y} & 0 \\ 0 & \frac{d\theta_z}{dF_y} & 0 & 0 & 0 & \frac{d\theta_z}{dM_z} \end{bmatrix} \tag{1}$$

Under an external load, **F**, the deformation of the single flexure hinge, **X**, can be expressed as:

$$\mathbf{X} = \mathbf{C}_{\text{Leaf\_spring}} \mathbf{F} \tag{2}$$

The external load and deformation of the single flexure hinge are defined in the following format, respectively:

$$\begin{aligned} \mathbf{F} &= \begin{bmatrix} F_x & F_y & F_z & M_x & M_y & M_z \end{bmatrix}^T \\ \mathbf{X} &= \begin{bmatrix} \Delta_x & \Delta_y & \Delta_z & \alpha_x & \alpha_y & \alpha_z \end{bmatrix}^T \end{aligned} \tag{3}$$

Figure 1 shows the design of the flexure hinge. To adapt to the profile of the continuum robot (i.e., round shape for high dexterity), the flexure hinges were designed to fit within concentric circles (outer radius: $r_{\text{out}}$, inner radius: $r_{\text{in}}$, respectively).

The 6-DoF compliance matrix of the single leaf spring, which was established in a convenient Cartesian coordinate system, cannot be conveniently used directly to obtain the compliance matrix of the flexure hinges. To solve this problem, a polar coordinate system was used to define the parameters of the flexure hinge. In this polar coordinate, the length variation of the flexure spring can be expressed as:

$$l(\theta) = r_{\text{in}} \sin \theta \tag{4}$$

where $\theta$ is the polar angle of the flexure hinge. By differentiating Equation (4), we can obtain the following expression:

$$dl(\theta) = r_{\text{in}} \cos \theta d\theta \tag{5}$$

The rotational compliance of the flexure hinge around the $z$ axis, $d\theta_z/dM_z$, can be expressed as:

$$\frac{d\theta_z}{dM_z} = \int_0^{\theta_m} \frac{r_{\text{in}} \cos \theta}{EI_z} d\theta = \frac{r_{\text{in}}}{EI_z} \int_0^{\theta_m} \cos \theta d\theta = \frac{r_{\text{in}} \sin \theta_m}{EI_z} \tag{6}$$

where $E$ is the elastic modulus of the flexure hinge; $\theta_m$ is the maximum polar angle of the leaf spring. $I_z$ is the second moment of area around the rotational axis of the given polar angle $\theta$, which can be expressed as:

$$I_z = \frac{hW^3}{12} \tag{7}$$

where $W$ is the width of the hinge.

In addition to the rotation generated by the moment $M_z$, the force $F_z$ can also contribute to the rotation of the flexure hinge, which can be expressed as:

$$\frac{d\theta_z}{dF_y} = -\int_0^{\theta_m} \frac{r_{\text{in}} \sin \theta r_{\text{in}} \cos \theta}{EI_y} d\theta = -\frac{r_{\text{in}}^2 \sin^2(\theta_m)}{2EI_y} \tag{8}$$

where $I_y$ is the cross-sectional area moment of inertia around the rotational axis of the given polar angle $\theta$, which can be expressed as:

$$I_y = \frac{h^3 W}{12} \tag{9}$$

The angular compliance of the flexure hinge about the $y$ axis, corresponding to the external moment $M_y$ can be expressed as:

$$\frac{d\theta_y}{dM_y} = \int_0^{\theta_m} \frac{r_{\text{in}} \cos\theta}{EI_y} d\theta = \frac{r_{\text{in}} \sin\theta_m}{EI_y} \tag{10}$$

Similarly, the external force $F_y$ will also contribute to the bending of the flexure hinge about the $y$ axis, which can be expressed as:

$$\frac{d\theta_y}{dF_z} = -\int_0^{\theta_m} \frac{r_{\text{in}} \sin\theta \, r_{\text{in}} \cos\theta}{EI_z} d\theta = -\frac{r_{\text{in}}^2 \sin^2(\theta_m)}{2EI_z} \tag{11}$$

The angular compliance about the $x$ axis can be expressed as:

$$\frac{d\theta_x}{dM_x} = \int_0^{\theta_m} \frac{r_{\text{in}} \cos\theta}{GJ_x} d\theta = \frac{r_{\text{in}} \sin\theta_m}{GJ_x} \tag{12}$$

where $G$ is the shear modulus of the material and $J_x$ is the polar moment of area of the cross-section.

It can be seen from Equation (1) that the compliance matrix of the flexure hinge is asymmetrical about its diagonal. The linear compliance along the $z$ axis corresponding to $M_y$ is the same as the rotational compliance around the $y$ axis corresponding to $F_z$; similarly, the linear compliance along the $y$ axis corresponding to $M_z$ is the same as the rotational compliance around the $z$ axis corresponding to $F_y$, which can be seen in Equations (8) and (11).

The linear compliance of the flexure hinge along the three axes (i.e., $x$, $y$ and $z$, respectively) can be expressed as:

$$\frac{dx}{dF_x} = \int_0^{\theta_m} \frac{r_{\text{in}} \cos\theta}{EA} d\theta = \frac{r_{\text{in}} \sin\theta_m}{EA} \tag{13}$$

$$\frac{dy}{dF_y} = -\int_0^{\theta_m} \left[ \int_0^{\theta} \frac{-r_{\text{in}} \sin\beta \, r_{\text{in}} \cos\beta}{EI_z} d\beta \right] d\theta = \frac{r_{\text{in}}^2}{4EI_z} \left( \theta_m - \frac{1}{2} \sin 2\theta_m \right) \tag{14}$$

$$\frac{dz}{dF_z} = -\int_0^{\theta_m} \left[ \int_0^{\theta} \frac{-r_{\text{in}} \sin\beta \, r_{\text{in}} \cos\beta}{EI_y} d\beta \right] d\theta = \frac{r_{\text{in}}^2}{4EI_y} \left( \theta_m - \frac{1}{2} \sin 2\theta_m \right) \tag{15}$$

where $A$ is the cross-sectional area.

Once the 6-DoF compliance matrix of a flexure hinge has been determined, the bending, torsion, axial loading and shear deformation of a beam can be obtained under the given external loads (i.e., force and moment). After that, the combined 6-DoF compliance matrix of the 2-DoF segment formed from several flexure hinges can be established by further extending Ryu's flexure mechanism modelling method.

### 2.2. Stiffness Model of a Flexure Hinge

In order to achieve the required mechanical performance (i.e., higher bending compliance and lower torsional compliance) of the continuum robot, the flexure hinge was designed to have a hexagonal shape, as seen in Figure 2. The compliance matrix of the flexure hinge can be established by considering the structural configuration of the leaf springs.

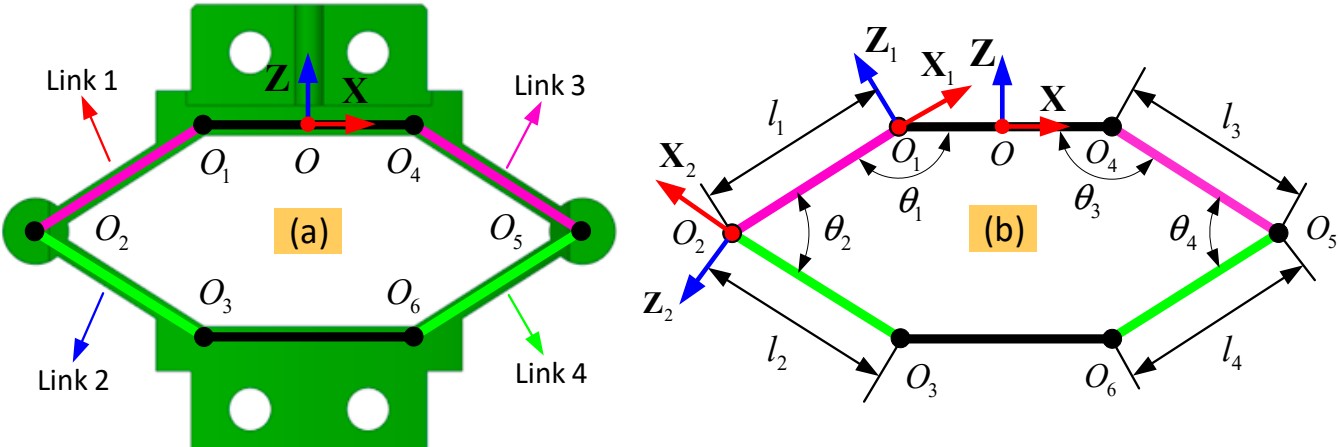

**Figure 2.** Geometry and parameter definition of the flexure hinge: (**a**) substructure configuration of the flexure hinge with serial links; (**b**) detail structures of the links and their configurations.

The flexure hinge is divided into four parts (i.e., Link 1 to Link 4) with their own local coordinate systems with the origins are attached at their ends ($O_1$ for Link 1, $O_2$ for Link 2, $O_4$ for Link 3 and $O_5$ for Link 4). To formulate the geometrical relationship of the links, the transformation matrix between the adjacent coordinate systems of the links is calculated. Taking Link 1 and Link 2 as the example, the included angle $\theta_1$ is used to calculate the transformation matrix.

Taking Link 1 as the example, the compliance matrix of the flexure hinge can be transformed from its local coordinate system $\{O_1\}$ to the global coordinate system $\{O\}$. The origin of the global coordinate system is located at the centre of line $O_1O_4$; the positive direction of **X** is along the vector $\overrightarrow{\mathbf{O_1O_4}}$; the positive direction of **Z** is in the plate of $O_1O_3O_4$ and vertical to the vector **X** in the upward direction; the vector **Y** is defined by the right-hand law. For the local coordinate system, it is defined by transforming the local coordinate system $\{O\}$ from $O$ to $O_1$, and then rotating around the **Y** axis with angle $-(\pi - \theta_1)$. Thus:

$$\mathbf{C}_O^{(1)} = \mathbf{T}_A \mathbf{C}_A^{(1)} \mathbf{T}_A^T \tag{16}$$

where $\mathbf{T}_A$ is the $6 \times 6$ transformation matrix, which can be defined in the following format:

$$\mathbf{T}_A = \begin{bmatrix} \mathbf{R} & \mathbf{0} \\ \mathbf{0} & \mathbf{R} \end{bmatrix} \begin{bmatrix} \mathbf{I} & \mathbf{\Phi}(r) \\ \mathbf{0} & \mathbf{I} \end{bmatrix} \tag{17}$$

$$\mathbf{\Phi}(r) = \begin{bmatrix} 0 & -r_z & r_y \\ r_z & 0 & -r_x \\ -r_y & r_x & 0 \end{bmatrix} \tag{18}$$

where **R** is the rotation matrix in the format of **ZYX** Euler angle; $\mathbf{\Phi}(r)$ is the skew-symmetric matrix defined by the vector $\begin{bmatrix} r_x & r_y & r_z \end{bmatrix}^T$ of the local coordinate system ($\mathbf{C}_A^{(1)}$) in the global coordinate system ($\mathbf{C}_O^{(1)}$); **I** is the ($6 \times 6$) identity matrix; $c\alpha = \cos\alpha$, $s\alpha = \sin\alpha$, $c\beta = \cos\beta$, $s\beta = \sin\beta$, $c\gamma = \cos\gamma$, $s\gamma = \sin\gamma$.

Based on the definition of the compliance matrix of each of the four links within the flexure hinge, the combined compliance matrix of the flexure hinge, $\mathbf{C}_O$, can be expressed as follows:

$$[\mathbf{C}_O] = \left( \left( \left[ \mathbf{C}_O^{(1)} \right] + \left[ \mathbf{C}_O^{(2)} \right] \right)^{-1} + \left( \left[ \mathbf{C}_O^{(3)} \right] + \left[ \mathbf{C}_O^{(4)} \right] \right)^{-1} \right)^{-1} \tag{19}$$

where $\mathbf{C}_O^1$, $\mathbf{C}_O^2$, $\mathbf{C}_O^3$ and $\mathbf{C}_O^4$ are the compliance matrices of the four links in the global coordinate system ($C_O$). It can be seen from the configuration of the flexure hinge that

Links 1 and 3 are combined in series with Links 2 and 4, respectively, and then they are further combined in parallel to form the overall structure of the flexure hinge.

The compliance matrix of the four links in Equation (19) can be expressed in their local coordinate systems by using the transformation matrixes.

$$
\begin{aligned}
\mathbf{C}_O^{(1)} &= \left[\mathbf{T}_O^A\right]^T \mathbf{C}_A^{(1)} \mathbf{T}_O^A \\
\mathbf{C}_O^{(2)} &= \left[\mathbf{T}_O^A \mathbf{T}_A^B\right]^T \mathbf{C}_B^{(2)} \mathbf{T}_O^A \mathbf{T}_A^B \\
\mathbf{C}_O^{(3)} &= \left[\mathbf{T}_O^D\right]^T \mathbf{C}_D^{(3)} \mathbf{T}_O^D \\
\mathbf{C}_O^{(4)} &= \left[\mathbf{T}_O^D \mathbf{T}_D^E\right]^T \mathbf{C}_E^{(4)} \mathbf{T}_O^D \mathbf{T}_D^E
\end{aligned}
\tag{20}
$$

where $\mathbf{C}_A^{(1)}$, $\mathbf{C}_B^{(2)}$ and $\mathbf{C}_D^{(3)}$ and $\mathbf{C}_E^{(4)}$ are the compliance matrices of the four links in their local coordinate systems, respectively, and $\mathbf{T}_O^A$, $\mathbf{T}_A^B$, $\mathbf{T}_O^D$ and $\mathbf{T}_D^E$ are the transformation matrices of the four links, based on Equations (17) and (18).

### 2.3. Stiffness Model of the Continuum Robot

#### 2.3.1. Load Calculation and Validation

After the compliance matrix of a single flexure hinge has been obtained, the compliance matrix of a 2-DoF segment, which is composed by combining the three identical parallel assembly flexure hinges (i.e., 120° interval), can be established. Taking one flexure hinge as an example, the local coordinate systems $\{\mathbf{C}_{i,j}\}$ and $\{\mathbf{C}_{i+1,j}\}$ are attached at the geometrical centre of the lower and upper platform, respectively, while another two coordinate systems $\{\mathbf{C}_i\}$ and $\{\mathbf{C}_{i+1}\}$ are attached at the centre of the lower and upper spatial joints, respectively, as seen in Figure 3.

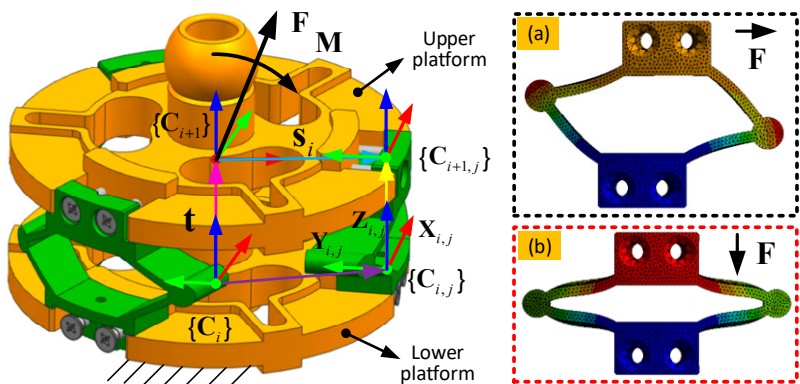

**Figure 3.** The coordinate systems definition for establishing the compliance matrix on an example segment: (**a**,**b**) are the simulated deformations of the flexure hinge under the given external loads (i.e., horizontal and vertical force, respectively).

As the segments are connected by central spatial, or ball, joints, the upper platform can only rotate about this joint. In order to define the rotational motion of the upper platform, the rotation vector, $\boldsymbol{n}$, is created defined by the **XOY** axes of the frame $\{\mathbf{C}_{i+1,j}\}$. When the upper platform is rotated about the vector, $\boldsymbol{n}$, for an angle, $\theta$, the rotation matrix, $\mathbf{R}_{C_{i+1}}$, of the upper platform can be expressed as:

$$
\mathbf{R}_{C_{i+1}} = \begin{bmatrix} k_x k_x vers\theta + c\theta & k_x k_y vers\theta - k_z s\theta & k_x k_z vers\theta + k_y s\theta \\ k_x k_y vers\theta + k_z s\theta & k_y k_y vers\theta + c\theta & k_y k_z vers\theta + k_x s\theta \\ k_x k_z vers\theta + k_y s\theta & k_y k_z vers\theta - k_x s\theta & k_z k_z vers\theta + c\theta \end{bmatrix}
\tag{21}
$$

where $s\theta = \sin\theta$, $c\theta = \cos\theta$, $vers\theta = 1 - \cos\theta$. $\theta$ is the rotation angle around the rotation axis. $k_x$, $k_y$, and $k_z$ are the three components of the rotation vector, $\boldsymbol{n}$, which can be expressed as $\boldsymbol{n} = \begin{bmatrix} k_x & k_y & k_z \end{bmatrix}^T$.

For a given rotation of the upper platform (defined by the rotation vector, $n$, and rotation angle $\theta$), the **ZYX** Euler angles of the flexure hinges around the three axes ($\Delta\theta$, $\Delta\psi$ and $\Delta\phi$, respectively) can be calculated using Gregory's method [31]. The **ZYX** Euler angles are expressed in the upper coordinate system $\{C_{i+1}\}$ but can be transferred to the coordinate system of the flexure hinge. Taking the $i$-th flexure hinge as an example, the displacement, $\Delta\mathbf{s}$, can be expressed as:

$$\Delta\mathbf{s} = \left(\mathbf{R}_{C_{i+1}} - \mathbf{R}_0\right)\mathbf{s}_i \tag{22}$$

where $\mathbf{R}_0$ is the initial posture of the upper platform; $\mathbf{s}_i$ is the position vector from the origin of $\{C_{i+1}\}$ to the origin of $\{C_{i+1,j}\}$.

Based on the established compliance matrix of the flexure hinge (Equation (19)), the load generated by the deformation can be expressed as:

$$\begin{bmatrix} \mathbf{f}_{C_{i,j}} \\ \mathbf{m}_{C_{i,j}} \end{bmatrix} = \left[\mathbf{C}_{i,j}\right]^{-1}\left[\mathbf{u}_{C_{i,j}}\right] \tag{23}$$

where $\mathbf{C}_{i,j}$ is the 6-DoF compliance matrix of the flexure hinge; $\mathbf{f}_{C_{i,j}}$ and $\mathbf{m}_{C_{i,j}}$ are the force and moment generated by the deformation of the 2-DoF segment; $\mathbf{u}_{C_{i,j}}$ is the displacement of the flexure hinge, which can be expressed as:

$$\mathbf{u}_{C_{i,j}} = \begin{bmatrix} \Delta\mathbf{s} & \Delta\theta \end{bmatrix}^T \tag{24}$$

The motion of the upper platform will result in the deformation of the three flexure hinges. Thus, the overall static balance equation for the segment can be expressed as:

$$\mathbf{M} = \sum_{j=1}^{3}\left(\mathbf{M}_{i+1,j} + \mathbf{F}_{i,j} \times \overrightarrow{O_{i+1}O_{i+1,j}}\right) \tag{25}$$

where $\mathbf{M}_{i,j}$ is the moment generated by the material deformation of the $i$-th flexure hinge; $\mathbf{F}_{i,j}$ is the force applied on the spatial joint of the lower platform; $\overrightarrow{O_{i+1}O_{i+1,j}}$ is the position vector from the origin of frame $\{C_{i+1}\}$ to the origin of frame $\{C_{i+1,j}\}$.

To validate the stiffness model, it was simulated in ANSYS Workbench (version: 2017). One 2-DoF segment (constructed by the upper and lower rigid disks, and three equally spaced flexible hinges) was modelled. The upper and lower rigid disks were set as rigid parts (deformation is assumed negligible) with the lower disk fixed, while the three flexible hinges were set as linear isotropic elastic materials (specifically, ABS as an example). Geometries and material properties are shown in Table 1. The comparisons between the theoretical calculation and simulation for the given external loads can be seen in Figure 4.

**Table 1.** Parameters of the leaf spring.

| Parameters | Explanation | Value | Unit |
|:---:|:---:|:---:|:---:|
| $h$ | Thickness of the flexure hinge | 1 | mm |
| $W$ | Width of the flexure hinge | 6 | mm |
| $l$ | Length of flexure hinge | 8 | mm |
| $r_{\text{in}}$ | Inner radius of flexure hinge | 17 | mm |
| $r_{\text{out}}$ | Outer radius of flexure hinge | 23 | mm |
| Material | Duramax, MAGNA | N/A | N/A |
| $E$ | Young's module of the material | $1.6 \times 10^3$ | MPa |
| $\sigma$ | Poisson's ratio of the material | 0.4 | MPa |

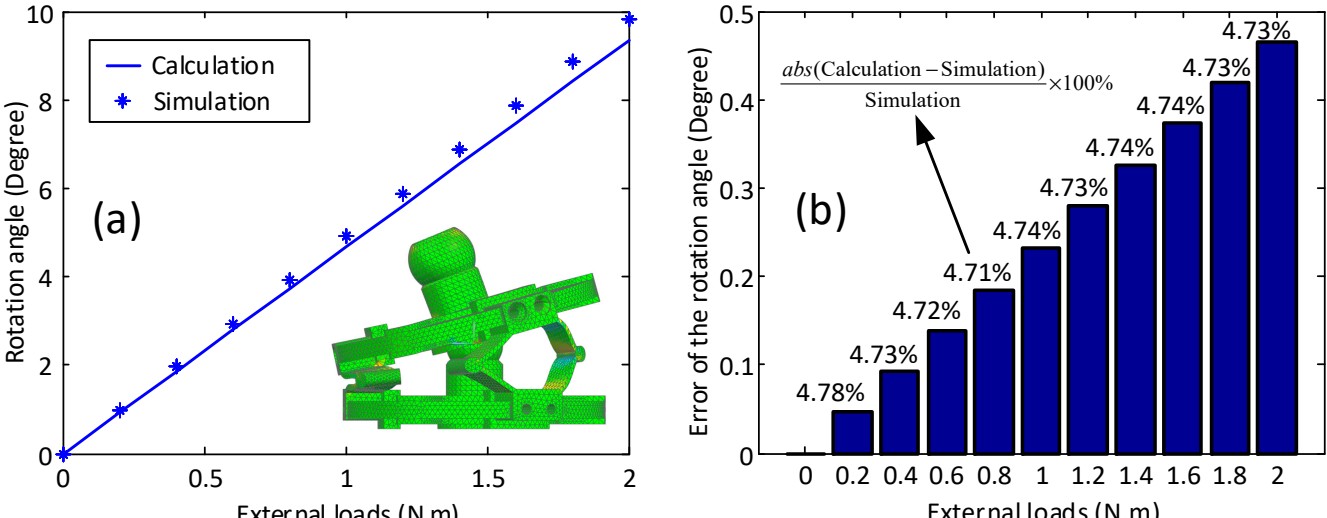

**Figure 4.** An example of the model validation on the 2-DoF segment: (**a**) is the rotation deviation under different external loads (phase angle, 180°); (**b**) is the deviation comparison between the calculation and simulation.

It can be seen here that the results from the calculation compare well with the simulation results, validating the developed model. Figure 4a illustrates the stiffness character of the proposed 2-DoF segment, which reflects the relationship between the deformation and the external loads. As can be seen, the flexure hinge design ensures a constant bending stiffness. Figure 4b shows the error between the proposed model and simulation at the test points. With the increase in the external load, the errors increase correspondingly (from 0.05° to 4.7° as the loads increase from 0.5 to 2 N·m). However, the relative error is almost constant at around 4.7%, indicating the developed model is reliable.

### 2.3.2. Stiffness Model of the Continuum Robot

Following the establishment of the kinetostatic model of the 2-DoF segment including the three flexure hinges, the full model of the 2-DoF segment, including driving cables, was developed to facilitate the establishment of the kinetostatic model of the full continuum robot, which is formed from the combination of multiple 2-DoF segments.

Figure 5 presents the schematic of the 2-DoF segment including the driving cables and flexure hinges. To describe the motion, the base platform was regarded as fixed, while the upper platform can achieve the 2-DoF rotation around the ball joint ('*A*'). Three coordinate systems (base coordinate system {*C*}, ball joint coordinate system {*A*} and upper platform ball joint {*O*}) are defined to derive the force and moment balance equations. The external loads (force, ***F***, and moment, ***M***) are applied at the upper platform, and cause the rotation of the upper platform around the ball joint *A*.

Thus, the static equation of the upper platform of the 2-DoF segment can be expressed as:

$$\begin{cases} \sum\limits_{i=1}^{3} \left( \mathbf{f}_{\text{cable},i} + \mathbf{f}_{\text{hinge},i} \right) + \mathbf{F}_{\text{joint}} = \mathbf{F} \\ \sum\limits_{i=1}^{3} \left\{ (\mathbf{b}_i + \mathbf{t}) \times \mathbf{f}_{\text{cable},i} + (\mathbf{c}_i + \mathbf{t}) \times \mathbf{f}_{\text{hinge},i} + \mathbf{M}_{\text{hinge},i} \right\} = \mathbf{M} \end{cases} \tag{26}$$

where $\mathbf{f}_{\text{cable},i}$ is the force in the i-th driving cable; $\mathbf{f}_{\text{hinge},i}$ and $\mathbf{M}_{\text{hinge},i}$ are the force and moment of the i-th flexible hinge, respectively; $\mathbf{F}_{\text{joint}}$ is the force on the ball joint; $\mathbf{b}_i$ and $\mathbf{c}_i$ are the position vectors from the origin of coordinate system {*C*} to the cable and flexible hinge attaching points, respectively; $\mathbf{t}$ is the position vector of the ball joint in the coordinate

system $\{C\}$; and $\mathbf{e}_i$ is the position vector of the upper cable attached position, which can be expressed as:

$$\mathbf{e}_i = \mathbf{b}_i + \mathbf{t} \tag{27}$$

Equation (26) can be written in the matrix form:

$$\mathbf{J}^T \mathbf{f} + \left[ \mathbf{C}_{\text{hinge}} \right]^{-1} \mathbf{s} + \mathbf{F}_{\text{joint}} = \mathbf{W} \tag{28}$$

where $\mathbf{W}$ is the wrench applied on the upper platform (expressed as: $\mathbf{W} = [\mathbf{F} \mathbf{M}]^T$), $\mathbf{f}$ is the force matrix of the three driving cables; $\mathbf{C}_{\text{hinge}}$ is the compliance matrix of the flexible hinge (from Equation (19)); and $\mathbf{s}$ is the rotational displacement of the 2-DoF segment.

The stiffness matrix of the 2-DoF segment can be established by taking the derivatives of the external load, $\mathbf{W}$, with respect to the rotational displacement, $\mathbf{s}$.

$$\mathbf{K}_{\text{segment}} = \frac{\partial \mathbf{W}}{\partial \mathbf{s}} = \mathbf{J}^T \frac{\partial \mathbf{f}}{\partial \mathbf{s}} + \frac{\partial}{\partial \mathbf{s}} \left( \mathbf{J}^T \right) \mathbf{f} + \left[ \mathbf{C}_{\text{hinge}} \right]^{-1} \frac{\partial \mathbf{s}}{\partial \mathbf{s}} = \mathbf{K}_1 + \mathbf{K}_2 + \left[ \mathbf{C}_{\text{hinge}} \right]^{-1} \tag{29}$$

The stiffness matrices $\mathbf{K}_1$ and $\mathbf{K}_2$ in Equation (29), which represent the structural stiffness and tendon stiffness, respectively, can be expressed as:

$$\mathbf{K}_1 = \mathbf{J}^T \frac{\partial \mathbf{f}}{\partial \mathbf{s}} = \mathbf{J}^T diag(k_1, k_2, k_3) \mathbf{J} \tag{30}$$

$$\mathbf{K}_2 = \frac{\partial}{\partial \mathbf{s}} \left( \mathbf{J}^T \right) \mathbf{f} = \sum_{i=1}^{n} \frac{f_i}{l_i} \begin{bmatrix} \mathbf{I} & [\mathbf{e}_i \times]^T \\ [\mathbf{e}_i \times] & [\mathbf{e}_i \times][\mathbf{e}_i \times]^T \end{bmatrix} + \sum_{i=1}^{n} f_i \begin{bmatrix} 0 & 0 \\ [\mathbf{n}_i \times]^T & -[\mathbf{n}_i \times][\mathbf{e}_i \times] \end{bmatrix} \tag{31}$$

where, $k_1$, $k_2$ and $k_3$ are the stiffness of the three driving cables; $\mathbf{J}$ is the Jacobian matrix of the 2-DoF segment; $f_i$ is the driving force of the $i$-th cable; $\mathbf{n}_i \times$ and $\mathbf{e}_i \times$ are the skew-symmetric matrix of the $\mathbf{n}_i$ and $\mathbf{e}_i$ vectors, respectively; and $\mathbf{n}_i$ is the unit vector of the i-th driving cable.

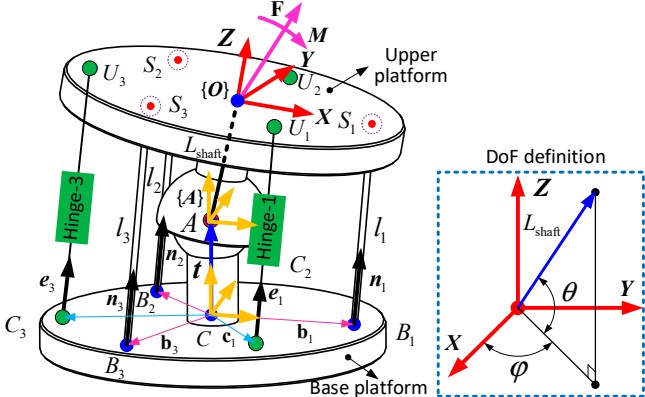

**Figure 5.** Static model of the 2-DoF segment constructed by the parallel-distributed driving cables and flexible hinges.

Further, due to the modular structure of the continuum robot (where each section is constructed by connecting several 2-DoF segments in series), the stiffness model of the $i$-th section can be expressed as:

$$\mathbf{K}_{\text{section},i} = \frac{\mathbf{K}_{\text{segment}}}{m_i} \tag{32}$$

where $m_i$ is the number of the segment for the $i$-th section.

As the 6 DoF continuum robot used here to validate the theory (see below) is constructed by serially connecting three sections made of 2-DoF segments, the stiffness model of the continuum robot, $\mathbf{K}_{\text{system}}$, can be expressed as:

$$\mathbf{K}_{\text{system}} = \left( \mathbf{K}_{\text{section},1}^{-1} + \mathbf{T}_{\text{section},2}\mathbf{K}_{\text{section},2}^{-1}\mathbf{T}_{\text{section},2}^{-1} + \mathbf{T}_{\text{section},3}\mathbf{K}_{\text{section},3}^{-1}\mathbf{T}_{\text{section},3}^{-1} \right)^{-1} \quad (33)$$

where $\mathbf{K}_{\text{section},1}$, $\mathbf{K}_{\text{section},2}$ and $\mathbf{K}_{\text{section},3}$ are the stiffnesses of each of the three sections (i.e., sections 1, 2 and 3 respectively), which can be calculated using Equation (32). $\mathbf{T}_{\text{section},2}$ and $\mathbf{K}_{\text{section},3}$ are the coordinate system transformation matrices of each section to the global coordinate system, respectively (see Equations (17) and (18)).

## 3. Model Validations and Performance Tests

In the previous section, the model of the flexure hinge (Equations (1)–(20)), 2-DoF segment (Equations (21)–(25)) and 6-DoF continuum robot (Equations (26)–(33)) were introduced. In order to validate the proposed kinetostatic model, a set of experiments were conducted by applying external loads to a prototype continuum robot. The displacements of the robot were determined from image analysis. Further, with the validated kinetostatic model, the performance of the continuum robot with varying external loads was investigated throughout its workspace to show its suitability for future applications.

### 3.1. Model Validation of Example 2-DoF Section

One section of the 6-DoF continuum robot (i.e., a 2-DoF continuum section formed by a set of modular 2-DoF segments), was selected as an example for validating the proposed kinetostatic model. Initially, the 2-DoF section was configured as a cantilever beam, with its base fixed while the displacement of the section under given external loads was measured, as seen in Figure 6. Grid paper (5 mm spacing) was utilized as the background for calibrating the shape of the section with a camera (to achieve the resolution of 0.2 mm in reality) placed in front of the test rig. The data obtained from the experiments are compared with the theoretical calculations.

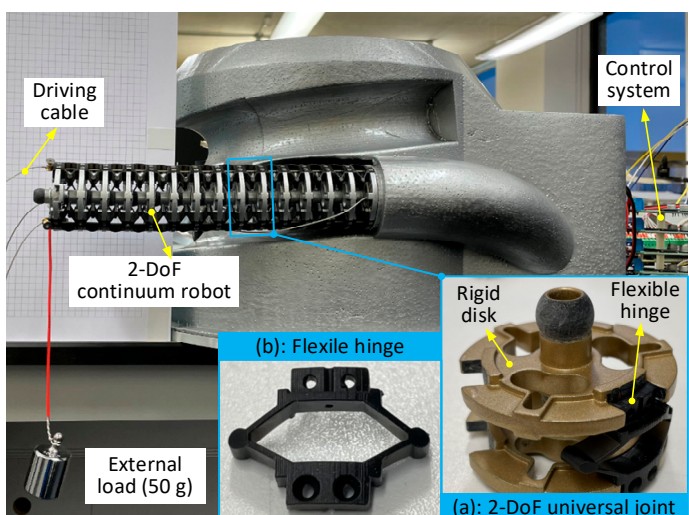

**Figure 6.** Experimental setup for validating the model for a continuum arm. A 2-DoF continuum robot was configured as a cantilever beam with an external load at the tip (length: 210 mm, outer diameter: 40 mm, number of disks: 15).

This section of the continuum arm is composed of 15 serially connected 2-DoF segments. The length of each segment is 15 mm; thus, the total length of the continuum arm is 210 mm. The physical components (i.e., rigid disk and flexure hinges, seen in Figure 1), were fabricated using additive manufacturing (Liquid Crystal Magna LCD Digital Light Processing) with different materials (i.e., rigid platforms: Photocentric Magna Hard; flexure

hinge: Photocentric Duramax) to achieve the desired mechanical performance. As the stiffness of the rigid disk is much higher than that of the flexure hinge, it is assumed that the deformation only occurs in the flexure hinges. With the external loads applied, the deformation in the flexure hinges results in rotation about the ball joint between the adjacent rigid platforms.

Actuation of the sections of the continuum robot is achieved by changing the length of flexible steel cables (surrounded by flexible spring tubes with a high compression stiffness to reduce friction and maintain curvature of the cables). There are three cables per section. The length of the cables is changed by spooling them about a pulley wheel connected to a geared DC motor housed in the base of the continuum robots with the control system (motor type: Maxon RE 25, gearbox: GP26 A with reduction ratio of 1:236, encoder: ENC16 with 1024 pulses). The modular closed-loop PID-based controllers for the motors were developed using an FPGA with a LabView interface (FPGA type: sbRIO-9627, LabVIEW version: 2017) and a speed controller for the motors (type: Pololu VNH5019), enabling a displacement resolution of 0.15 μm for the driving cables. This control system allows for the shape, or configuration, of the continuum robot to be controlled in real-time using position closed-loop control. For this experiment, the motors were first turned to remove the slack in the cables and then controlled to maintain a static position.

With the 2-DoF continuum section in a cantilever beam configuration, weights with different masses (20, 50, 100, and 200 g) were hung at the end of the section. The displacement of the section with and without the flexure hinges, as determined via image analysis, is seen in Figure 7.

As expected, Figure 7a shows the deformation of the 2-DoF continuum section with the flexure hinges increasing monotonically with load (i.e., from 0.6 to 8.2 mm as the load increases from 20 to 100 g. In Figure 7b, the equivalent deformation of the 2-DoF section without the flexure hinges is seen to be much higher than with the flexure hinges (tip displacement with flexure hinges: 0.6, 1.7, 3.6 and 8.2 mm, displacement without flexure hinges: 30, 67.5, 100.4 and 118.6 mm, respectively). This corresponds to an improvement in the average stiffness by a factor of 32.8, by adding the flexure hinges to the continuum robot.

Figure 7c illustrates the errors between the theoretical calculations (Equation (32) multiplied by the wrench caused by the external load) and experimental results for the tip displacement of the 2-DoF continuum section under load (from Figure 7a). Overall, the simulation results can match those from the experiments at the measured points with the average error of 9.1% (the percentage errors at the external loads 20, 50, 100, and 200 g are 14.8%, 10.8%, 6.1%, and 4.9%, respectively), which further validates the developed model.

With the kinetostatic model of the 2-DoF continuum section validated, the predicted mechanical performance of the section throughout its workspace was investigated. Specifically, the two permissible rotations (phase angle and bending angle) were varied throughout their allowable ranges to change the configuration of the 2-DoF continuum section, before the external load (200 g) was applied at the tip to determine the expected deviation from the set position, as shown in Figure 8. The workspace of the 2-DoF continuum section forms a paraboloid, with the position deviation map featuring a similar shape. It can be seen in Figure 8 that the position changes more with the bending angle than the phase angle, which means that the deviation between the set position and real position under load should be expected to increase when the bending angle is increased.

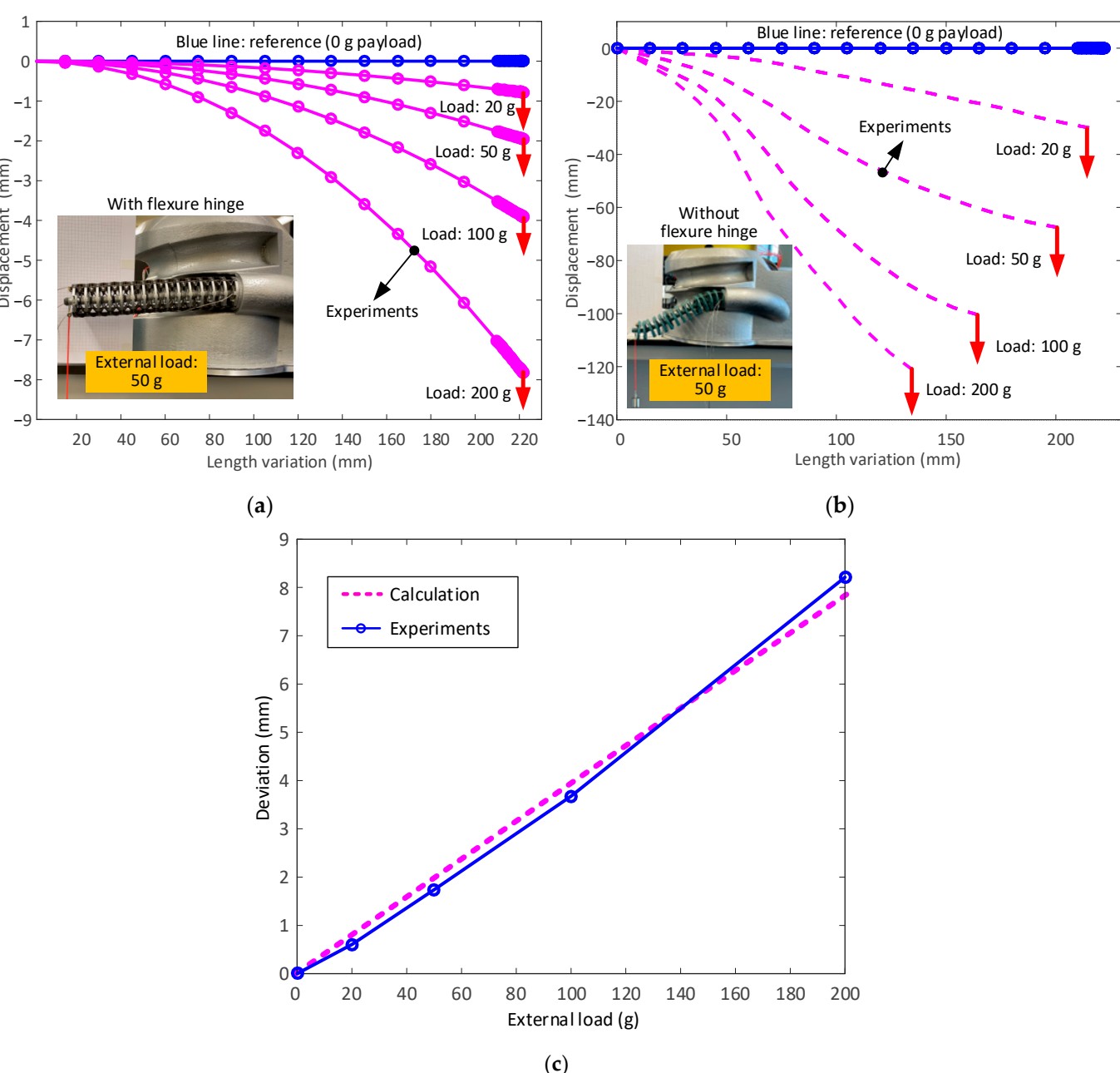

**Figure 7.** Deformation of the 2-DoF continuum robot under external loads: (**a**,**b**) are the deviation of the 2-DoF continuum robot with and without the flexure hinges; (**c**) is the comparison of the deviation between the theoretical calculation and experimental tests for the 2-DoF continuum robot with flexure hinges. Note: the arrows in (**a**,**b**) are the load directions.

### 3.2. Model Validation on the 6-DoF Continuum Robot

The 6-DoF continuum robot is formed of three 2 DoF sections. As a result, the kinetostatic character of the 6-DoF continuum robot will be very different than that of the single 2 DoF continuum section. Here, the developed control system was extended to allow the kinetostatic performance of the 6-DoF continuum robot to be tested under different configurations and external loads, enabling the evaluation of its mechanical performance (e.g., positional accuracy with payload) for use in practical applications.

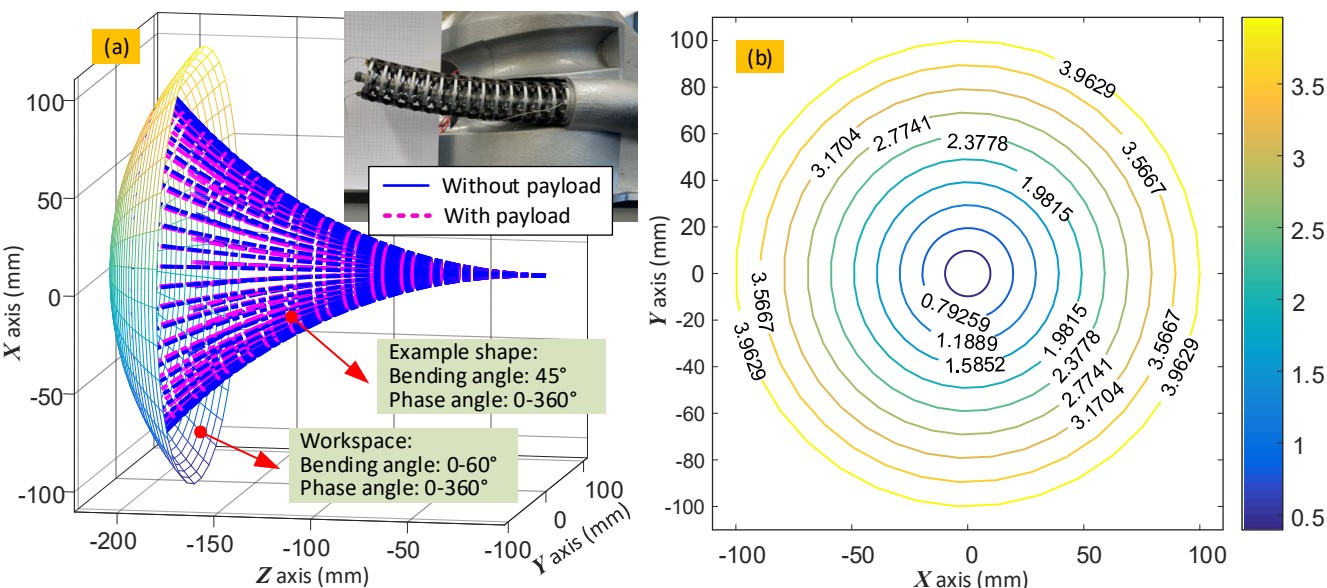

**Figure 8.** Position deviation of the 2-DoF continuum arm under the given external loads within the workspace: (**a**) position deviation of the continuum robot with bending angle varying from $0°$ to $360°$ and phase angle varying from $0°$ to $360°$; (**b**) counter map of the position deviation within the workspace. Note: the constant external load of 200 g is applied at the tip of the 2-DoF continuum arm as the case study.

To implement the desired shape control of the 6-DoF continuum robot (i.e., phase angle $\varphi$ and bending angle $\theta$ of the three 2-DoF sections), nine motors were controlled using an extended version of the real-time position closed-loop control system described above. To achieve the desired configuration, the required lengths of the nine driving cables were determined using a constant curvature approximation [9]. This represents the desired shape of the robot under no load. The motors were then actuated to set the cable lengths in the robot and then controlled to maintain their position. Loads, in the form of simple weights, were then applied to the tip, or end effector, of the robot to check the stiffness performance of the system, as shown in Figure 9. The deviation of the actual shape of the continuum robot under load compared to the set configuration was determined in a similar fashion to that described in the previous section.

The initial shape of the 6-DoF continuum arm was configured using the closed-loop control system to set the phase angle and bending angles for each of the three sections as $20°$ and $0°$, respectively. Figure 10a shows the shape variation of the 6-DoF continuum robot under the given external loads. As with the single section, the deviation from the set position increases with the external load applied at the tip. As the 6-DoF continuum arm operates in a cantilever configuration (i.e., fixed at the base), and is composed of the three sections, thus increasing the length of the robot threefold, the stiffness of the system is inherently much lower than that of a single section. As such, the tip displacement increases from 4 mm when the lighter external load (i.e., 20 g) is applied to around 35 mm when the external load is 200 g.

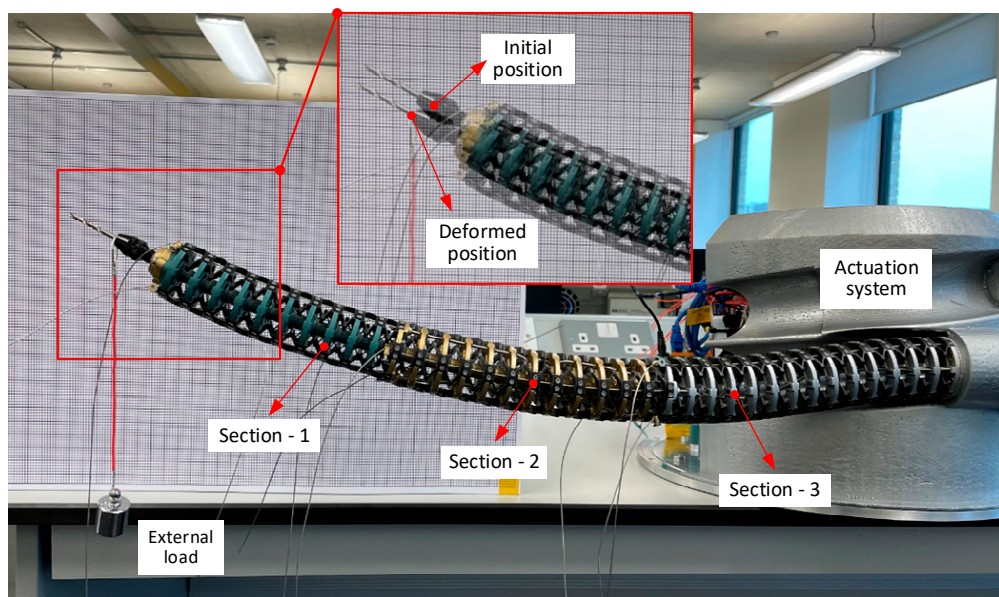

**Figure 9.** Experimental setup for testing the stiffness performance of the 6-DoF continuum robot. Configuration of the robot was set and maintained using nine closed-loop controllers to control the length of the driving cables. The external load was applied at the tip, while the corresponding change in position was captured by image analysis.

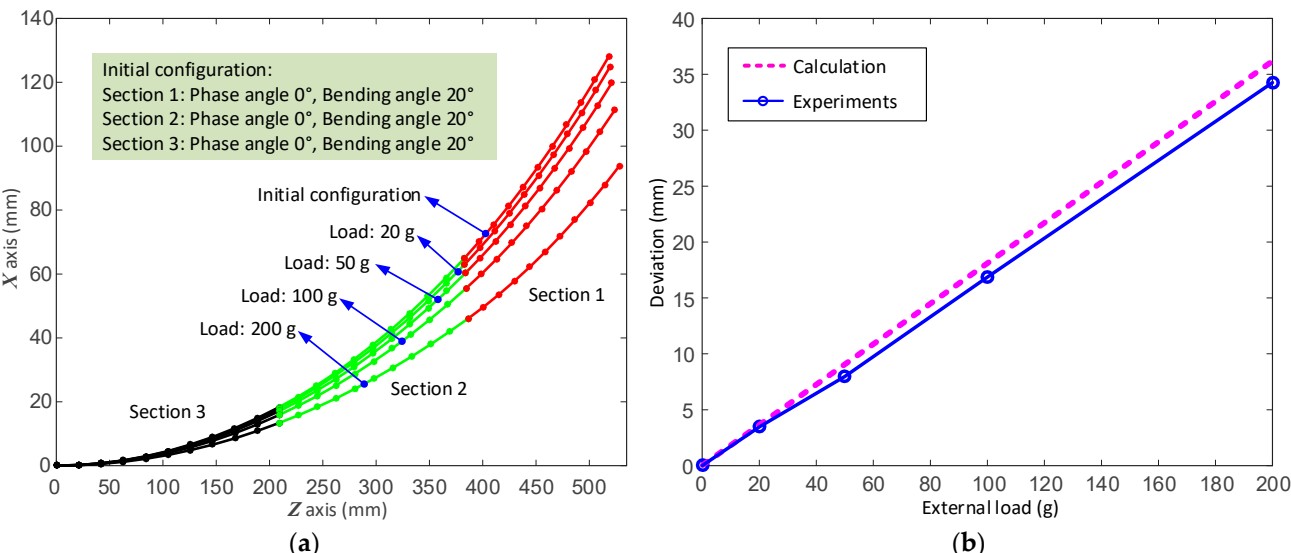

**Figure 10.** Comparison between the modelled and experimental results of the deflections on 6-DoF continuum robot under different loads: (**a**) is the deviation of the 6-DoF continuum robot under different loads; (**b**) is the results comparison between calculation and test.

Figure 10b illustrates the errors between the theoretical calculations (derived from Equation (33)) and experimental tip displacement data (from Figure 10a) for the 6-DoF continuum arm in the same configuration as a function of load. Overall, the theoretical calculations can match the experimental results under the tested external loads with the overall average error of 2.3% (i.e., the errors at the external loads 20, 50, 100 and 200 g are 5.4%, 9.1%, 4.5% and 5.5%, respectively). With the increase in the external load, the absolute error between the calculation and test increases (i.e., 0.2, 0.8, 0.9 and 2 mm, respectively) but the relative error decreases. Deviations between the theory and experimental data can be assumed to be mostly due to difficult-to-measure deviations in the parameters given in Table 1, due to manufacturing tolerances. The errors seem to be systematic and can be reduced through calibration.

With the developed model for the 6-DoF continuum robot validated for some given poses, the stiffness character of the system throughout its workspace can be studied. As the slender continuum arm is always in a cantilever beam configuration during standard operation, the stiffness of the system may become low in configurations with large bending and phase angles. Figure 11 shows the deviation in the position of the 6-DoF continuum arm throughout the workspace under a 200 g external load. In a similar fashion to the results from a single section, Figure 11a shows the shape of the 6-DoF continuum arm is a paraboloid shape (when all sections are set to the same angles) and remains in a similar shape when the load is applied. Figure 11b displays the counter map of the deviation in the tip displacement of the 6-DoF continuum arm under load. Similar to Figure 10b, the deviations are circular, which means that there is little change in position when the robot is moving about the central axis (i.e., with the bending angle unchanged, and the phase angle changed from 0° to 360°). However, there are larger deviations when the bending angle is increased.

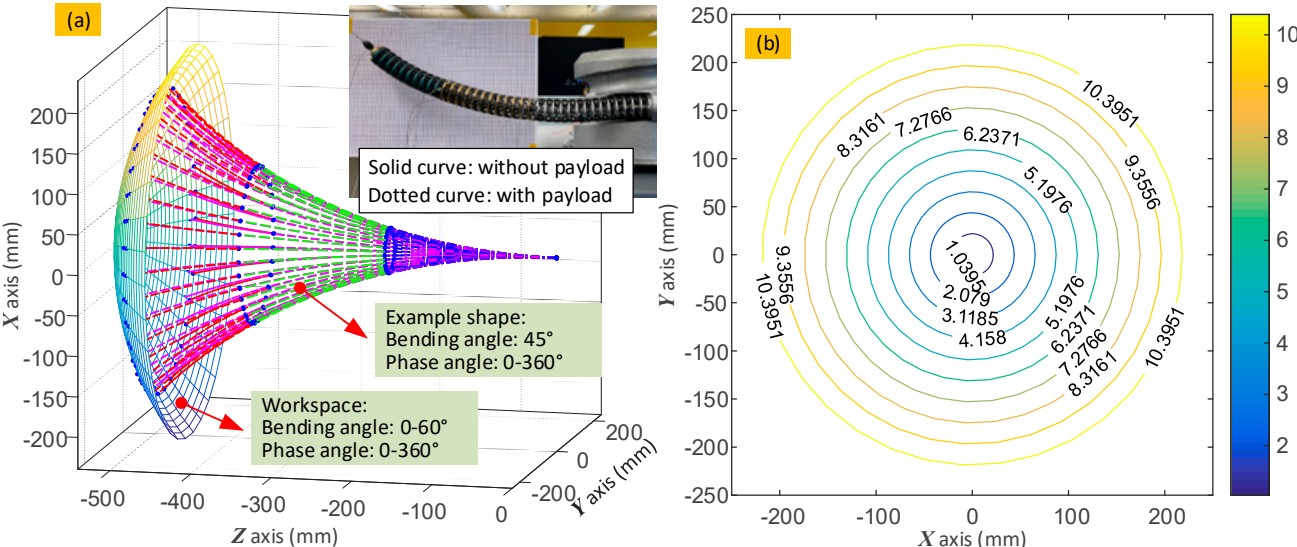

**Figure 11.** Position deviation of the 6-DoF continuum arm under the given external loads within the workspace: (**a**) position deviation of the continuum robot with bending angle varying from 0° to 360° and phase angle varying from 0° to 360°; (**b**) counter map of the position deviation within the workspace. Note: the constant external load of 200 g is applied at the tip of the 2-DoF continuum robot as the case study.

In this section, with the aid of the developed prototype and control system of the continuum robot, the kinematic model developed in previous sections was validated, and the performance of the continuum robot with the flexible hinges included shown to be beneficial. It can be seen the developed model compared favourably with experiments with a positional accuracy of 9.1% and 6.2% for the 2-DoF and 6-DoF continuum arms, respectively. Furthermore, the ability of the 6-DoF continuum robot to assume different configurations was tested by implementing the given trajectory with the aid of a real-time controller. It was shown in the experimental results that the kinematic accuracy of the system can be significantly improved (by a factor of 38.2) with the inclusion of flexure hinges. The results demonstrate the developed 6-DoF continuum robot is an effective tool with potential for performing remote operations in hazardous environments with high accuracy (10 mm accuracy within a ±250 mm workspace with 200 g external payload) and demonstrates significant robustness.

## 4. Conclusions

In this paper, a novel continuum robot formed with serially connected modular 2-DoF backbone structures was developed with the potential to perform complex tasks with high accuracy even under load.

It was shown that the kinematic performance of conventional high length–diameter ratio backbone-structured continuum robots can be improved through the inclusion of additional compliant elements separate from the backbone itself. As an example, a flexure hinge-based design was selected for constructing a 6-DoF continuum robot.

A new kinematic and stiffness model of the proposed 6-DoF continuum robot, which incorporates the model of a modular cable-driven 2-DoF parallel mechanism and the novel flexure hinges, was established. With the developed kinematic model of the 6-DoF continuum robot, the configuration of the continuum robot (bending angle and phase angle of each section) can be regulated to perform a given task. Through these models, which were validated through experiment and simulation, it was also found that the introduction of the flexure hinge significantly improves the stiffness of the 6-DoF continuum robot.

In addition, for the continuum robot prototype and associated real-time control system, the kinematic accuracy of the system was also significantly improved by adding the flexure hinges. It was found that the kinematic accuracy of the continuum robot can be improved by a factor of 32.8 with the aid of the flexure hinges. Using the validated model, the stiffness behaviours of the 2-DoF and 6-DoF continuum arms were tested throughout their respective workspaces. The results demonstrate that the developed continuum arm can achieve high accuracy operation with the aid of the flexure hinges (i.e., 2-DoF continuum arm: 4 mm accuracy throughout a 110 mm workspace; 6-DoF continuum arm: 10 mm accuracy throughout a 250 mm workspace).

**Author Contributions:** Conceptualization, N.M., D.C. and S.M.; methodology, D.C.; software, N.M.; validation, N.M., D.C.; formal analysis, D.C.; investigation, N.M.; resources, D.C. and S.M.; data curation, N.M.; writing—original draft preparation, N.M.; writing—review and editing, D.C.; visualization, D.C.; supervision, D.C.; project administration, D.C.; funding acquisition, D.C. and S.M. All authors have read and agreed to the published version of the manuscript.

**Funding:** This research was funded by [EPSRC] grant number [EP/V027379/1] And the APC was funded by [EP/V027379/1].

**Institutional Review Board Statement:** Not applicable.

**Informed Consent Statement:** Not applicable.

**Data Availability Statement:** Not applicable.

**Conflicts of Interest:** The authors declare no conflict of interest.

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
