# Peer review of "Modelling and Analysis of the Spital Branched Flexure-Hinge Adjustable-Stiffness Continuum Robot"

_robotics, doi:10.3390/robotics11050097_

Round 1
Reviewer 1 Report
The study proposes a new flexible robot with a modular structure that adjusts the stiffness.
The exact location of Section C-C is not clear.
What does lowercase w(θ) represent and is it really variable?
What sort of dimensions, parameters and materials were entered in the ANSYS simulations? Are the results consistant with that direction the direction of the load?
It is not clear how a 0.2mm resolution can be achieved with the experimental protocol described. Since a simple camera is used, how does the perspective effect will affect the measurement?
Since a the grid paper does not cover the origin of the continuous manipulator, how is it calibrated?
The use of outer structure improve the performance is suggested as opposition to inner structure, but the author do not explain its advantages compared to the latter (in terms of design, accuracy, etc.)
Author Response
The study proposes a new flexible robot with a modular structure that adjusts the stiffness.
The exact location of Section C-C is not clear.
Authors: Thank you for your comments. The location of section C-C is at the top surface of the flexible hinge. The new figure has been updated in the paper to show it. Please check the figure.1 at page 3.
What does lowercase w(θ) represent and is it really variable?
Authors: The w(θ) is the thickness variation of the flexible hinge. It is a variable with relation to the angle θ. When θ is zero at the initial stage, w(θ) is the wall thickness of the ring. However, with the increase of the angle θ, the thickness w(θ) is increasing gradually.
What sort of dimensions, parameters and materials were entered in the ANSYS simulations? Are the results consistent with that direction the direction of the load?
Authors: The 2-DoF segment is selected from the base section of the 6-DoF arm. The detail dimensions and parameters of the flexible hinge of the 2-DoF segment can be seen from Table.1. The classical ABS material was selected in the ANSYS software for the model validation.
Yes, the results shown in Figure.4 are same with the deformation direction of the FEM results. The material used in the simulation has been added in the paper, please check the highlighted part at page 8.
It is not clear how a 0.2mm resolution can be achieved with the experimental protocol described. Since a simple camera is used, how does the perspective effect will affect the measurement?
Authors: The grid paper with 5 mm spacing is placed as the background for scaling the deformation of the deviation of the continuum arm by a camera placed in front. For each set of experiment, the figures were captured by the camera and saved for the further analysis by the additional software (i.e., ScanIt.exe). As the camera has high pixel, the deviation can be sliced into the pixels. By checking the number of pixels in the software with the calibration process, the actual deviation of the continuum arm can be obtained. With the certain distance between the camera and experimental setup, the 0.2 mm resolution can be achieved.
Since the grid paper does not cover the origin of the continuous manipulator, how is it calibrated
Authors: It is true that the grid paper does not cover all the continuum arm in the experimental setup. As the camera is parallelly placed in front of the experimental setup, the deviations of the origin of the continuum arm that not covered by the grid paper can also be predicted after calibrating the system. As the pixels are used to determine the deviation of the continuum arm, even there is no grid paper, the pixels are still available.
The use of outer structure improves the performance is suggested as opposition to inner structure, but the author do not explain its advantages compared to the latter (in terms of design, accuracy, etc.)
Authors: This new structure of continuum arm (i.e., the additional modular flexible hinges for adjusting the performance of system) is designed based on the previous versions. In previous, the backbone structure (e.g., a NiTi rod with same dimension from base to tip, compressional springs) is normally adopted to design the continuum arm, which has the following disadvantages: 1) performance of the system (e.g., stiffness) cannot be regulated after prototyping; 2) as the torsional stiffness of NiTi rod and compressional springs are normally lower, the torsional stiffness of the continuum arm around its axis is low; 3) due to the working character of the driving cables (can only provide pull force, not the push force), the torsional stiffness is uncontrollable.
However, by using the outer structure (flexible hinge), the structures can be designed in particular shape to have the desired performances, for example: 1) the diamond shape of the flexible hinge is designed to improve the torsional stiffness of the system; 2) thickness of the flexible hinges in base section (close to the actuation system) is higher than the tip section, improving the stiffness of the base section.
One example of the previous continuum arm with backbone structure can be found in this reference [1], where an additional “stiffness adjustable mechanism” is used to adjust stiffness of the system but with complicated operation process and bulk dimension. However, by using the concept developed in this paper, the aforementioned challenges are not existing anymore. Further, please check the highlighted part at page 2 and 3.
[1] Ma, Nan. "Design and modelling of a continuum robot with soft stiffness-adjustable elements for confined environments." (2022).

Reviewer 2 Report
Some sentences need to be rewritten for better presentation. Examples are as follows:
Further, a robotic joint in which the stiffness can be actively controlled was developed 69 by using multiple rotary flexure hinges [19]
Please rewrite
However, if flexure hinges are to be integrated, the kinematic behaviour of the overall system needs to be studied further [25].
Please rewrite
-The contributions of the presented paper should be written explicitly in the introduction
-The references should be updated to include one or more 2022 references
Author Response
Some sentences need to be rewritten for better presentation. Examples are as follows:
Further, a robotic joint in which the stiffness can be actively controlled was developed 69 by using multiple rotary flexure hinges [19]. Please rewrite.
Authors: This sentence has been rewritten. Please check the highted part at page 2.
However, if flexure hinges are to be integrated, the kinematic behaviour of the overall system needs to be studied further [25]. Please rewrite.
Authors: This sentence has been rewritten. Please check the highted part at page 2.
-The contributions of the presented paper should be written explicitly in the introduction
Authors: The contribution of this paper has been addressed in the introduction section. Please check the highlighted part at page 3.
-The references should be updated to include one or more 2022 references
Authors: Some more up-to-date references have been added in the paper, please check the highlight references at page 18.

Reviewer 3 Report
In this paper a comprehensive mathematical model of the flexure hinge, 2-DoF segment and 6-DoF continuum robot were introduced for a continuum robot. The kinetostatic model was validated through experiment and simulation, it was also found that the introduction of the flexure hinge improves the stiffness of the 6-DoF continuum robot significantly.
Congratulations to the authors, who are presenting an elaborated and correct mathematical model. It will be very intersting to such a robot, how it will perform in industrial environment.
I have some remarks to the editing of the paper. It seems that the different authors have performed differt tasks (this is a good thing :) ), but there are some corrections to make for the paper, to be unitary edited.
Line 33: instead of “and shape”, to be more clear, should stay “and robot shape”
Line 118: for logical reason instead of “disk i+1 and i”, should stay “disk i and i+1”
Figure 1: Please indicate, that the “Flexible hinge” (magenta frame) is reverse presented as in the mechanism on the left side of the figure. The view on the right side of the figure is actually the top view of the hinge, not any section view. In this case, the small red frame inside the magenta frame is obsolete.
Line 123-124: Please refer to the h parameter depicted in Figure 1, and please present the l and θm parameter in Figure 1.
Please move the “where ?? is the maximum polar angle of the leaf spring.” Right after Eq.(6).
In Figure 2, a coordinate system is used for the hinge. The origin of the coordinate system it seems to be in the same O point, but the axes are not the same as in Figure 1. Please use the same denotations. From topology point of view of the flexible hinge, actually there are two chains in the link (O1O2O3 and O4O5O6) consisting each of them by two links (ex. O1O2 and O2O3). That for the denotation of Link1, Link2, … is more appropriate as the Chain1, Chan2, … .
Because there is already the xOy coordinate system presented (in color), there is no need to present it again in the lower left corner of the figure.
Please reformulate the sentence from line 201-203. For now, in my understanding, the X axes are alongside the geometrical features. I do not understand the meaning of ?1.
Line 204-205: Please define the global and local coordinate systems.
In the Figure 3 there is the third variant of the coordinate system used for the flexure hinge!! Please use for the flexible hinge the same coordinate system, with the same notations through the paper!
In Figure 5. the position of coordinate system {A} will be more understandable if graphically would be represented between the two gray platforms.
Because the ground forces are not taken into account in eq.(26), line 307 should be referred to the static equation of the upper platform, not the whole 2-DoF segment (which includes the base platform also, as the upper paragraph describes).
In Figure 1. the thickness of the leaf-spring is denoted by h, but in Table 1. is referred as parameter t.
Line 403: the load is increased to 200g based on the Figure 7(a)
In Firgure 8. the labels a) and b) are missing. The analogy of the photo and graph will be maximized if the Z axis of the graph would be horizontal in Fig.8(a), as in Fig.11(a).
To be in accordance with Fig.8(a), the X axis from Fig.10(a) should be axis Z.
In Figure 11(a), please use the same style of legend as in Fig.8(a).
Author Response
In this paper a comprehensive mathematical model of the flexure hinge, 2-DoF segment and 6-DoF continuum robot were introduced for a continuum robot. The kinetostatic model was validated through experiment and simulation, it was also found that the introduction of the flexure hinge improves the stiffness of the 6-DoF continuum robot significantly.
Congratulations to the authors, who are presenting an elaborated and correct mathematical model. It will be very intersting to such a robot, how it will perform in industrial environment.
I have some remarks to the editing of the paper. It seems that the different authors have performed differt tasks (this is a good thing :) ), but there are some corrections to make for the paper, to be unitary edited.
Line 33: instead of “and shape”, to be more clear, should stay “and robot shape”
Authors: The new words have been added in the paper. Please check the highlighted part at page 1.
Line 118: for logical reason instead of “disk i+1 and i”, should stay “disk i and i+1”
Authors: It has been corrected in the paper. Please check the highlighted part at page 3.
Figure 1: Please indicate, that the “Flexible hinge” (magenta frame) is reverse presented as in the mechanism on the left side of the figure. The view on the right side of the figure is actually the top view of the hinge, not any section view. In this case, the small red frame inside the magenta frame is obsolete.
Authors: The section view has been changed to top view. Please check figure.1. Further, the notes have been added in the caption of Figure.1. Please check the highlighted part in the caption area of Figure.1.
Line 123-124: Please refer to the h parameter depicted in Figure 1, and please present the l and θm parameter in Figure 1.
Authors: The parameter h has been explained and referred in the main text. Please check the highlighted part at page 3. L and θm have been presented in the figure.1.
Please move the “where ?? is the maximum polar angle of the leaf spring.” Right after Eq.(6).
Authors: It has been moved after Eq.(6). Please check the highlighted part at page 4.
In Figure 2, a coordinate system is used for the hinge. The origin of the coordinate system it seems to be in the same O point, but the axes are not the same as in Figure 1. Please use the same denotations. From topology point of view of the flexible hinge, actually there are two chains in the link (O1O2O3 and O4O5O6) consisting each of them by two links (ex. O1O2 and O2O3). That for the denotation of Link1, Link2, … is more appropriate as the Chain1, Chan2, … .
Authors: The definitions (i.e., denotation directions) of the coordinate system in Figure.2 has been corrected same with Figure.1. The “Chain” has been replaced to “Link” to make it consistent with the topology expression.
Because there is already the xOy coordinate system presented (in color), there is no need to present it again in the lower left corner of the figure.
Authors: The coordinate system in left corner has been removed.
Please reformulate the sentence from line 201-203. For now, in my understanding, the X axes are alongside the geometrical features. I do not understand the meaning of ?1.
Authors: This sentence has been rewritten to make it clear. Please check the highlighted part at page 6.
Line 204-205: Please define the global and local coordinate systems.
Authors: The definition of global and local coordinate systems have been added in the paper, please check the highlighted part at page 6.
In the Figure 3 there is the third variant of the coordinate system used for the flexure hinge!! Please use for the flexible hinge the same coordinate system, with the same notations through the paper!
Authors: The direction of the coordinate systems have been corrected to be same with the previous definitions. Please check figure.3 at page 7.
In Figure 5. the position of coordinate system {A} will be more understandable if graphically would be represented between the two gray platforms.
Authors: The graphically features have been added in figure.5 to make the coordinate system of {A} more understandable. Please check the new Figure.5 at page 9.
Because the ground forces are not taken into account in eq.(26), line 307 should be referred to the static equation of the upper platform, not the whole 2-DoF segment (which includes the base platform also, as the upper paragraph describes).
Authors: The new definition of the Eq.(26) has been updated in the paper, which can be checked at the highlighted part at page 9.
In Figure 1. the thickness of the leaf-spring is denoted by h, but in Table 1. is referred as parameter t.
Authors: The editing error has been corrected in the paper. t has been corrected to h in table.1.
Line 403: the load is increased to 200g based on the Figure 7(a)
Authors: The editing error has been corrected in the paper. please check the highlighted part at paper 13.
In Firgure 8. the labels a) and b) are missing. The analogy of the photo and graph will be maximized if the Z axis of the graph would be horizontal in Fig.8(a), as in Fig.11(a).
Authors: The missing labels of (a) and (b) have been added. The direction of the graph has been adjusted to be same with the photo. Please check the figure.8 and figure.11.
To be in accordance with Fig.8(a), the X axis from Fig.10(a) should be axis Z.
Authors: The name of the axis has been corrected based on the figure.8.
In Figure 11(a), please use the same style of legend as in Fig.8(a).
Authors: The legend of Figure.11 has been amended. As three colours were used to plot the three sections of the continuum arm. The simulation and calculation were categorized into the solid and dotted curves for the legend. Please check the figure.11.

Round 2
Reviewer 1 Report
The authors have addressed all the reviewer's comments.